# Noise characteristics in spaceflight multichannel EEG

**Patrique Fiedler** [1]*, **Jens Haueisen**[1], **Ana M. Cebolla Alvarez**[2], **Guy Cheron** [2], **Pablo Cuesta**[3], **Fernando Maestú**[3], **Michael Funke**[4]

1 Institute of Biomedical Engineering and Informatics, Technische Universität Ilmenau, Ilmenau, Germany,
2 Université Libre de Bruxelles, Brussels, Belgium, 3 Complutense University of Madrid, Madrid, Spain,
4 McGovern Medical School, University of Texas, Houston, Texas, United States of America

* patrique.fiedler@tu-ilmenau.de

## Abstract

The cognitive performance of the crew has a major impact on mission safety and success in space flight. Monitoring of cognitive performance during long-duration space flight therefore is of paramount importance and can be performed using compact state-of-the-art mobile EEG. However, signal quality of EEG may be compromised due to the vicinity to various electronic devices and constant movements. We compare noise characteristics between in-flight extraterrestrial microgravity and ground-level terrestrial electroencephalography (EEG) recordings. EEG data recordings from either aboard International Space Station (ISS) or on earth's surface, utilizing three EEG amplifiers and two electrode types, were compared. In-flight recordings showed noise level of an order of magnitude lower when compared to pre- and post-flight ground-level recordings with the same EEG system. Noise levels between ground-level recordings with actively shielded cables, and in-flight recordings without shielded cables, were similar. Furthermore, noise level characteristics of shielded ground-level EEG recordings, using wet and dry electrodes, and in-flight EEG recordings were similar. Actively shielded mobile dry EEG systems will support neuroscientific research and neurocognitive monitoring during spaceflight, especially during long-duration space missions.

## Introduction

Space missions critically depend on the cognitive, sensory-motor and emotional performance of the spacecraft's crew. Adverse effects can occur during space flights [1–5], especially for long-duration space missions [6], making early detection and countermeasures implementation highly relevant. Neurocognitive assessment of crew members to detect possible cognitive and behavioral conditions which might pose a risk for the mission is essential [7–10].

Cognitive performance during long-duration missions may be influenced by several factors such as microgravity and radiation in space [11–13]. The altered vestibular input can disturb the dynamic balance of visual and somatosensory systems which control functions such as posture, eye-hand coordination, spatial orientation, and navigation [2, 14]. Other factors,

**Data Availability Statement:** All relevant data will be available from the Laboratory of Neurophysiology and Movement Biomechanics at Université Libre de Bruxelles upon request and after approval from the European Space Agency

Medical Board (ESA-MB) and the NASA Johnson Space Centre Institutional Review Board (NASA-IRB). Authors do not own the data but others can request data from ESA and NASA. The authors confirm that they did not have any special access or request privileges that others would not have.

**Funding:** This work was supported in part by the German Federal Ministry of Education and Research (BMBF) grant TeleBrain (01DS19009A, JH), the Free State of Thuringia within the ThiMEDOP project (2018 IZN 0004, JH and PF) with funds of the European Union (EFRE), the Belgian Federal Science Policy Office and the European Space Agency (ESA) (AO2004, 118, GC and AMC), and the European Union's Horizon 2020 research and innovation program under a Marie Skłodowska-Curie grant (101007521, JH, PF, and FM). The funders had no role in study design, data collection and analysis, decision to publish, or preparation of the manuscript.

**Competing interests:** The authors declare no competing or conflicting interests.

including personal-social experience in microgravity, can influence cognitive and affective changes [5]. Previous studies often used surrogate environments and model systems for testing [15–17]. Due to the limited transferability of results from these surrogate systems, research on changes in brain function during actual space flight is important.

Both further brain research, and periodical assessment will contribute to increase our understanding of the underlying adaptive functional mechanisms. Periodical assessment will allow assessment, feedback and adjustment of an astronauts' ongoing performance. The respective investigations cannot be performed in surrogate environments but need to be performed during ongoing and future space missions. Besides the indirect assessment of brain function via behavioral testing, direct neurocognitive assessment is highly desirable. Most monitoring devices are relatively heavy and thus currently not suitable for space flight, while EEG is uniquely suited due to its lightweight, compact, and mobile character. Moreover, EEG allows real-time monitoring of neuronal activity with high temporal resolution [18, 19]. Recent advances both in terms of sensors and electronics support the self-application of multichannel EEG with significantly reduced preparation times [20–22] which enable frequent EEG measurements during space missions. Moreover, these systems can be worn by moving subjects [18, 22] and do not require electrolyte gels, pastes, or extensive cleaning agents thanks to the dry electrodes.

Besides general challenges of EEG recordings such as the accessibility of the sources of the signals of interest and the considerable intra- and inter-individual variability in the recorded signals, several challenges are specifically relevant during space missions. Most importantly, EEG data is affected by technical and biological interferences. Stationary and transient interferences, referred to as noise and artifacts, overlay the EEG signals and often exhibit higher amplitudes than the signal of interest, necessitating sophisticated signal preprocessing [23, 24]. Technical interferences may include mechanical (e.g., vibrations), chemical (e.g., electrolyte stability), and electromagnetic disturbances (e.g., power supplies). They are caused by interaction with other (medical) devices. The coupling mechanism can be galvanic, inductive, capacitive, or electromagnetic waves. In terms of electromagnetic compatibility, EEG devices should fulfill the CE or similar standards, minimizing such interaction. Biological interferences include all other biosignals except the desired EEG. Interferences originating from sweating or movement (e.g., respiration, blood pulsation, swallowing) can have a considerable influence on the EEG. For example, movements can cause disturbances of the electrical double layer at the electrodes and variations of the polarization potential, skin stretch artifacts, and additionally, changes in the coupled technical interferences.

No systematic reports are available on the existence, nature, and extent of these interferences in in-flight extraterrestrial microgravity environments. Successful EEG studies have been performed and published on different aspects of the physiological and psychological impact of spaceflight conditions [1–14]. However, skepticisms about the EEG signal quality and interferences remain. While the technical EEG recording equipment aboard ISS has been verified and validated by experts, no analysis dedicated to the EEG signal quality is publicly available till now. Acquisition of specific in-flight condition EEG recordings on astronaut populations for the unique purpose of investigating signal quality and interferences seems impractical given the limited available resources while facing extensive associated costs and efforts. Consequently, the present dedicated technical analysis on existing, previously recorded EEG data in Space is of utmost relevance to provide a publicly available basis for planning future studies both in terms of methodological and hardware considerations.

We aim to compare the noise characteristics of in-flight recordings to pre-flight, post-flight, and other ground-level EEG recordings. Based on the visual inspection of the few previously published EEG traces and spectra [25–28], we investigated three primary hypotheses:

1. The noise characteristics aboard ISS are not considerably different from ground-level recordings of the same setup.

2. Recording setups including active shielding technology lead to lower noise levels.

3. Dry electrode recordings on earth may provide similar noise characteristics when compared to in-flight data using gel-based electrodes.

Because no data recorded specifically for interference analysis are available, we analyzed previously recorded data from the multi-electrode electroencephalogram mapping module (MEEMM) [5] of the European physiology module installed in the Columbus segment of the ISS. We compared them to ground-level recordings with the same device. Since a newer and more advanced EEG system (eego system, ANT Neuro B.V.) with dry electrodes and active shielding is currently being used in the Shenzhou 13 mission on the Chinese TIANHE space station, we included ground-level recordings of this system in the comparison.

## Methods

### EEG datasets

We compared six datasets of resting-state EEG recorded either aboard the International Space Station (ISS) or on terrestrial ground-level, including overall three EEG amplifier models and two electrode types.

Three datasets have been recorded using the MEEMM, a fully stationary installation supporting DC-EEG measurements with up to 128 channels using unshielded cables. The MEEMM uses a dedicated physical reference electrode. The three datasets include pre-flight sessions (15 recordings, performed on ground-level), in-flight sessions (10 recordings, performed aboard ISS), and post-flight sessions (6 recordings, performed on ground-level). An extended 10–20 layout, comprising 59 gel-based EEG electrodes, was used for all recordings.

An additional set of 14 post-flight recordings was acquired using an asalab 64-channel amplifier (ANT Neuro B.V., Hengelo, Netherlands) on ground-level in a standard lab environment. The asalab amplifier is a stationary DC-EEG amplifier with a common average reference. The electrodes were connected to the amplifier using micro-coaxial cables for active shielding. During the recordings, an extended 10–20 layout, comprising 61 gel-based EEG electrodes, was used.

All four aforementioned datasets have been acquired on five ISS crew members in the frame of a cognitive EEG study paradigm involving sessions of 2 minutes of resting-state EEG with open eyes and 2 minutes of resting-state EEG with closed eyes [4, 5]. Electrode-skin impedances were checked prior to the recordings using the respective measurement functions of the MEEMM and asalab amplifier. For all electrodes, the impedance was ensured to be below a threshold of 5 kOhm.

The resting-state EEG recordings of Cebolla et al. 2016 [4] were compared to similar recordings performed on ground-level using the eego system (ANT Neuro B.V.) in a standard lab environment. The eego amplifier is a mobile DC-EEG amplifier with dedicated physical reference. The electrodes were connected to the amplifier using micro-coaxial cables supporting active shielding. Two datasets corresponding to two types of EEG caps have been compared: a conventional gel-based EEG cap and a novel Multipin dry electrode cap [21]. Both datasets comprised 30 resting-state EEG recordings with closed and open eyes, respectively. The caps comprised 256 electrodes in a quasi-equidistant layout. Electrode-skin impedances were checked prior to the recordings using the integrated function of the eego amplifier. For all gel-based recordings, a maximum threshold of 50 kΩ for 90% of the channels was defined. For the dry electrode recordings, no threshold was defined [21].

**Table 1. EEG dataset properties.**

| Dataset no. | Amplifier | Condition | Electrode type, shielding | No. of recordings | Sampling rate | EEG Reference | Study |
|---|---|---|---|---|---|---|---|
| 1 | MEEMM | ground-level (pre-flight) | gel-based, unshielded | 15 | 1116 | Right earlobe | [4, 5] |
| 2 | MEEMM | aboard ISS (in-flight) | gel-based, unshielded | 10 | 1116 | Right earlobe | |
| 3 | MEEMM | ground-level (post-flight) | gel-based, unshielded | 6 | 1116 | Right earlobe | |
| 4 | asalab | ground-level (post-flight) | gel-based, active shielding | 14 | 1024 | Common average | |
| 5 | eego | ground-level | gel-based, active shielding | 30 | 1024 | Vertex | [21] |
| 6 | eego | ground-level | dry Multipin, active shielding | 30 | 1024 | Right mastoid | |

A summary of the amplifier types and technical parameters of the compared recordings is provided in Table 1.

All data from studies of Cebolla et al. 2016 [4] and Fiedler et al. 2022 [21] have been acquired in accordance with relevant guidelines, regulations, and the ethical standards outlined in the Declaration of Helsinki. All volunteers provided written informed consent. The study of Cebolla et al. 2016 was approved by the European Space Agency Medical Care Committee and the NASA Johnson Space Centre Institutional Review Board for Human Testing [4]. The study of Fiedler et al. 2022 was approved by the Ethics commission at the medical faculty of the Friedrich-Schiller-University Jena, Germany [21].

In order to minimize the impact of differences in signal metrics caused by mental state, body movements, or eye movements, we only analyzed comparable data epochs across all conditions. All analyzed data were recorded under resting condition with either open or closed eyes. During in-flight conditions, body movements and muscle activity were minimized by means of specific body fixation belts fixated on the Columbus segment walls close to the MEEMM recording module. During all ground-level recordings, volunteers were sitting in a relaxed position on a comfortable chair. During all recordings with open eyes, the volunteers were asked to focus their eyes on a dedicated fixation spot.

## Data processing and analysis

Comparing dataset 2, recorded aboard ISS, to the different ground-level recordings, we investigated the three hypotheses defined above. Given the hypotheses and the differing recording setups, we aimed to homogenize the datasets prior to comparison. We, extracted a subset of 55 channel positions from the extended ten-twenty electrode layouts present in all datasets 1 to 4. A topographic plot of the extracted electrode subset is shown in Fig 1. For datasets 5 and 6 we extracted the 55 electrode positions of the equidistant electrode layout with minimum Euclidean distance to the aforementioned 10–20 positions.

Mobile recordings are intended to be performed during future long-duration spaceflight missions in parallel to regular activities of the spacecraft's crew. Mobile EEG often focuses on AC-EEG due to unavoidable movement-induced large amplitude low frequency artifacts hampering DC-EEG analysis. We, therefore, focused our investigation on AC-EEG in the frequency band (0,100] Hz and therefore subtracted the initial channel offsets from the data. Subsequently, analysis sequences of 30 seconds duration were selected within the 2 minutes recordings. The data segments were specifically selected by EEG experts, avoiding body movement, eye movement, or eye blinking artifacts in the analyzed data.

Power spectral density (PSD) for all data sequences was calculated using the Welch estimation method. Bad channels in datasets 1–4 were automatically identified by evaluating the mean PSD of a given channel in the frequency band [70,100] Hz. According to Eq (1), a given channel i is identified as a bad channel if its PSD is higher than the mean PSD + threefold

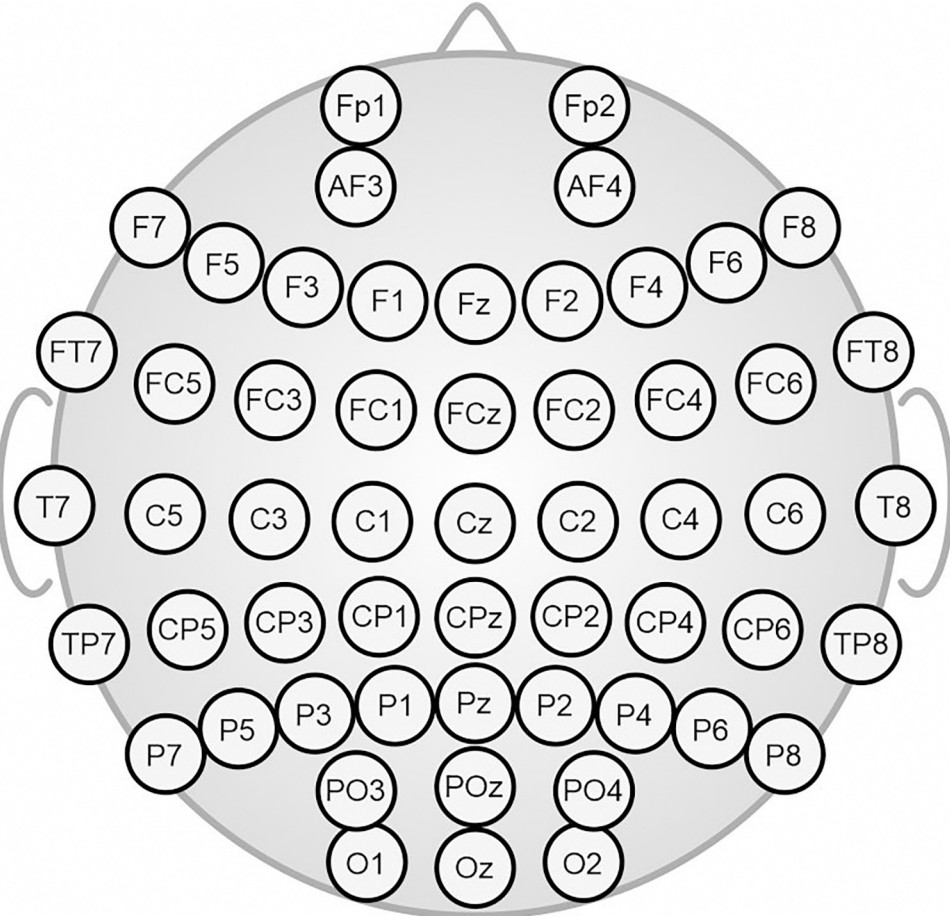

**Fig 1. Extended 10–20 subset comprising 55 channels contained in and extracted from datasets 1–4.** For datasets 5 and 6, 55 electrode positions of the 256-channel equidistant electrode arrangement with minimum Euclidean distance to the shown 10–20 positions have been analyzed.

standard deviation of all channels of a given dataset/condition.

$$PSD_i > \overline{PSD} + 3\sigma_{PSD} \qquad (1)$$

Bad channels in datasets 5 and 6 were defined following Fiedler et al. 2022 [21]. All bad channels were excluded from further processing and analysis. The remaining channels were re-referenced to common average reference. Finally, the PSDs were recalculated based on the re-referenced data excluding bad channels. No resampling or filtering was applied to the data.

## Results & discussion

### In-flight vs. ground-level recordings

Comparing averaged PSDs of EEG signals recorded using identical amplifier and cap systems enables comparison of the general noise level impact on the signals.

For MEEMM, bad channels during pre-flight (open eyes: 0.8%; eye-closed: 0.6%), in-flight (open eyes: 0.9%; eye-closed: 1.1%), and post-flight (open eyes: 1.2%; closed eyes: 1.2%) were identified. Excluding bad channels, the PSDs of all three recording conditions using the unshielded MEEMM system are shown in Fig 2.

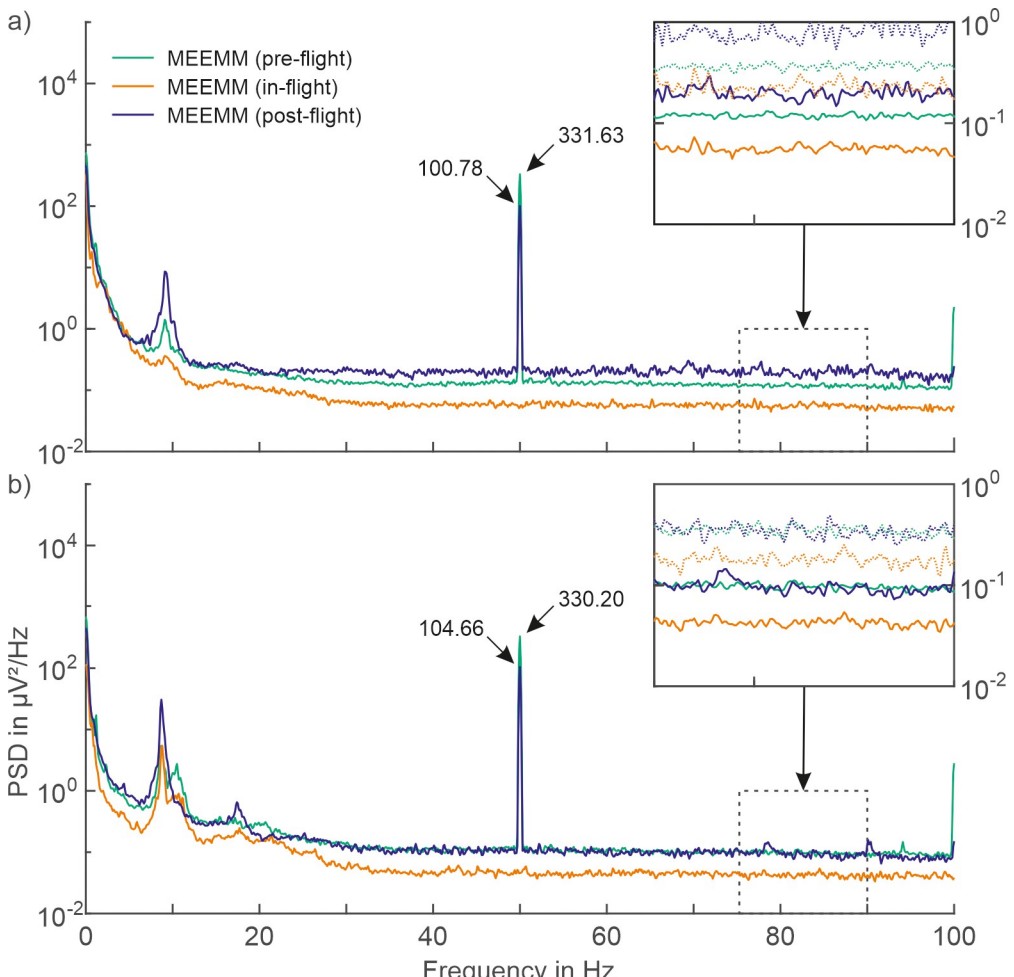

**Fig 2. Comparison of power spectral density (PSD) between in-flight and ground-level EEG recordings.** Average PSDs of 30 seconds of resting-state EEG data recorded with the MEEMM system during ground-level (pre- and post-flight) and in-flight conditions with a) open eyes, and b) closed eyes of the participants. Solid lines represent mean; dotted lines represent mean + standard deviation.

The PSD is increased for ground-level recordings for frequencies above 5 Hz with open eyes, and 2 Hz with closed eyes, compared to the corresponding in-flight recordings. The highest noise level is evident for the post-flight recordings during resting-state EEG recorded with open eyes. However, the increased means for both ground-level recordings (pre-flight, post-flight) are below the standard deviation of the in-flight recordings for the majority of the investigated frequencies. Given similar electrode-skin impedances levels ($< 5$ kOhm), as well as the preceding bad-channel and artifact exclusion for all three compared conditions, the observed differences in the PSD levels can not be caused by impedance differences or bad channels.

Both ground-level recordings of the MEEMM system exhibit considerable powerline interference at 50 Hz for the pre-flight (331.63 $\mu V^2$/Hz for open eyes; 330.20 $\mu V^2$/Hz during closed eyes) and post-flight (100.78 $\mu V^2$/Hz for open eyes; 104.66 $\mu V^2$/Hz during eye-closed) conditions. In contrast, no considerable powerline interference is evident in the in-flight recordings.

The PSD during in-flight condition is almost an order of magnitude lower for most frequencies when compared to pre- and post-flight conditions. This may, considering the unshielded wires, indicate a lower environmental noise level during the in-flight condition. Even though the

distance of the participants to various electronic components in the vicinity of the recording setup inside the ISS may be low, the design and housing of these devices may ensure a low environmental noise level. Furthermore, the primary power supply within the ISS is battery DC power [29, 30], which further explains the absence of 50 Hz or 60 Hz powerline interference.

## Active shielding

Active shielding and active electrodes are two methods used to reduce environmental noise coupling in EEG recordings [31, 32]. While devices using either method have already been used during in-flight EEG [4–6, 26, 27, 33], active shielding may be preferred for high-density EEG and long-duration spaceflight due to reduced complexity of cabling and weight. Fig 3 illustrates the impact of active shielding on the PSDs performed at ground-level with two different amplifier systems (stationary asalab; mobile eego system), excluding bad channels.

In the asalab ground-level recordings, 1.6% and 2.3% of the channels have been classified as bad channels during recordings with open eyes and closed eyes, respectively. For the eego gel-based recordings, 8.9% and 6.4% of the channels were identified as bad channels during open eyes and closed eyes, respectively.

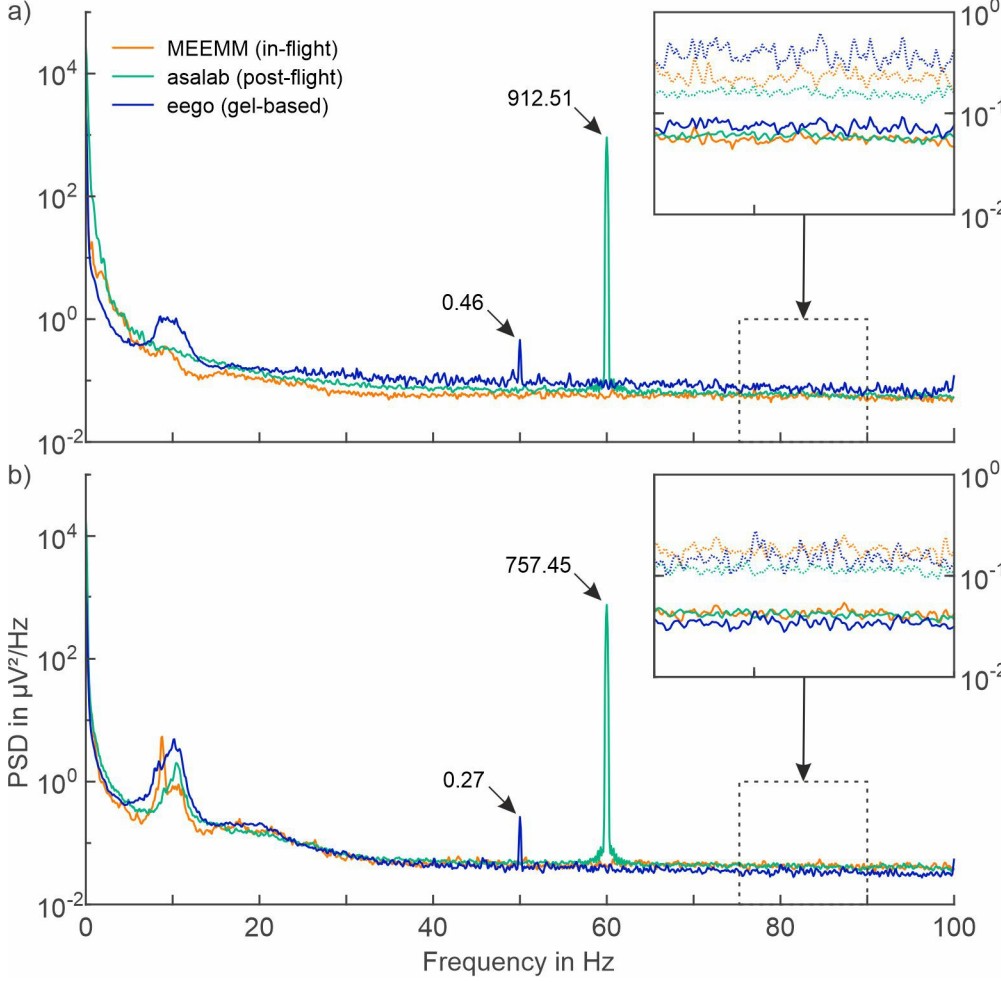

**Fig 3. Comparison of power spectral density (PSD) between unshielded and actively shielded EEG recordings.** Average PSDs of 30 seconds of resting-state EEG data recorded with unshielded (MEEMM, in-flight), and actively shielded gel-based electrode caps (asalab / eego, ground-level) conditions with a) open eyes, and b) closed eyes of the participants. Solid lines represent mean; dotted lines represent mean + standard deviation.

Even though recordings have been performed in greatly varying environments (in-flight vs. ground-level) of differing sample size Cebolla et al. 2016 [4] vs. Fiedler et al. 2022 [21], the general characteristics of the PSDs of all three recording conditions are similar both in terms of overlapping mean PSD and standard deviation.

Increased low-frequency power in the interval (0,7) Hz during eye-opened is more evident with MEEMM and asalab recordings when compared to eego recordings. This observation may be attributed primarily to eye movements. No considerable difference in the aforementioned frequency range is found during recordings with closed eyes.

A low level of powerline interference at 50 Hz (0.46 $\mu V^2$/Hz during open eyes; 0.27 $\mu V^2$/Hz during closed eyes) is evident for ground-level recordings performed with the eego amplifier. Considerable powerline interference at 60 Hz (912.51 $\mu V^2$/Hz during open eyes; 757,45 $\mu V^2$/Hz during closed eyes) is visible in ground-level recordings using the asalab amplifier. The increased powerline interference of the post-flight recordings using the asalab amplifier may be attributed to the measurement environment, i.e. increased number or proximity of powerlines or power supplies during the recording.

Comparison between the PSD of actively shielded ground-level recordings to unshielded in-flight recordings illustrates the equivalent signal characteristics and the effectiveness of the active shielding method in reducing environmental noise compared to the unshielded ground-level recordings of the MEEMM system (cp. Fig 2). While active shielding may not further improve the signal quality evident during the in-flight recordings, it may support more constant signal quality during varying recording environments, a crucial requirement for long-term, repetitive and mobile measurements during long-duration spaceflight missions.

## Dry electrodes

Gel-based electrodes pose considerable limitations for in-flight EEG given the need for transporting consumables (gels, cleaning agents, disinfectants), in addition to technical and health-related risks. Moreover, gel-based systems require expert preparations and cannot be self-applied. Dry electrodes solve the aforementioned issues and provide a convenient alternative for in-flight EEG monitoring during long-duration spaceflight.

Dry electrodes have previously been shown to provide signal quality comparable to gel-based EEG during laboratory [20, 21], and mobile conditions [22]. Dry electrode ground-level recordings showed 18.7% and 17.4% bad channels during open eyes and closed eyes, respectively. Fig 4 shows a comparison of the PSDs of gel-based and dry ground-level EEG recordings performed using the eego amplifier and the gel-based in-flight MEEMM recordings, excluding bad channels. No considerable differences in the PSD's characteristics can be identified, with both means and standard deviations of the three conditions strongly overlapping.

Dry electrode recordings showed increased powerline interference (18.00 $\mu V^2$/Hz during open eyes; 15.29 $\mu V^2$/Hz during closed eyes) when compared to gel-based electrodes using the same eego amplifier at ground-level. However, it is noteworthy that the powerline interference with dry electrodes using the eego amplifier remained considerably lower than both gel-based asalab and MEEMM ground-level recordings.

The comparison of gel-based and dry recordings using the same amplifier provides evidence for equivalent signal characteristics like the MEEMM in-flight recordings, and therefore underlines the potential of dry electrodes to be used for EEG monitoring during long-duration spaceflight missions. Our results provide evidence that equivalent signal quality for dry and gel-based systems can be expected in the in-flight recording environment.

While an electrode-skin impedance threshold of 5 kOhm was defined for the MEEMM in-flight recordings (cp. Methods), the gel-based recordings using the eego amplifier were

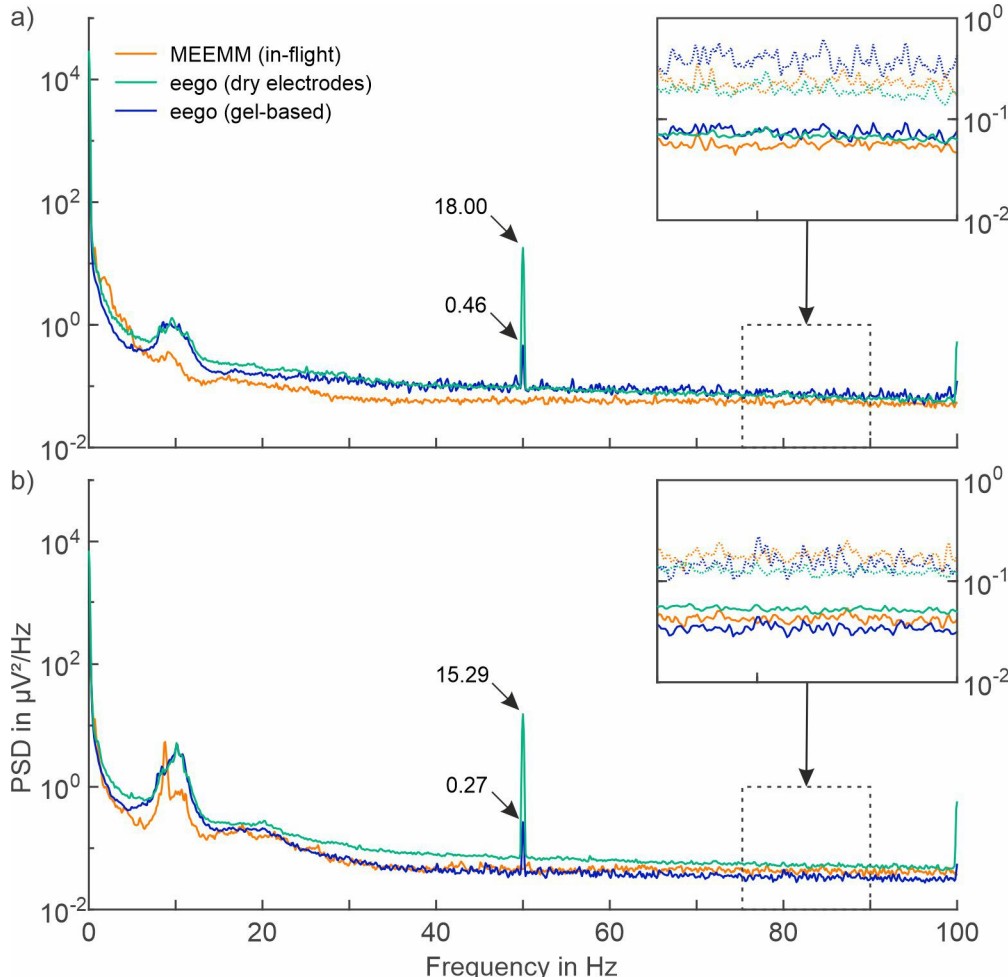

**Fig 4. Comparison of power spectral density (PSD) between gel-based and dry EEG recordings.** Average PSDs of 30 seconds of resting-state EEG data recorded with unshielded (MEEMM, in-flight), and actively shielded gel-based and dry electrode caps (eego, ground-level) conditions with a) open eyes, and b) closed eyes of the participants. Solid lines represent mean; dotted lines represent mean + standard deviation.

performed with a threshold of 50 kOhm for 90% of the channels. The mean and standard deviation of the electrode skin impedances for the gel-based recordings with the eego amplifier were 24 ± 18 kOhm. No impedance threshold was defined for the dry electrode recordings. The mean and standard deviation for the dry electrode recordings were 532 ± 199 kOhm. However, as shown in previous publications, when using state-of-the-art EEG amplifiers with very high input impedances and supporting noise reduction methods like active shielding, the electrode-skin impedance level no longer has a strong impact on signal quality [34–36]. As shown in Fiedler et al. 2022, electrode-skin impedances of the dry Multipin electrodes do not show a strong correlation with channel reliability up to 900 kOhm [21].

## Summary

Although previously published studies on neurophysiological and psychological effects of microgravity conditions using EEG have proven evidence that EEG in space successfully asses brain function, literature lacks a dedicated analysis of the noise levels of EEG recorded during

spaceflight. Consequently, concerns about environmental electromagnetic noise levels in spacecraft remained, which must be addressed to allow objective assessment of hardware and software requirements for long-term EEG during future long-distance spaceflight missions.

We analyzed datasets for six different recording conditions and compared spectral characteristics and signal noise levels of EEG recorded aboard ISS and on ground-level. Constraints for our analyses include a) the use of available previously recorded and published data given the limitations impeding performing dedicated onboard recordings; b) the low number of volunteers in previous in-flight EEG studies limiting statistical analysis. We selected an analysis approach avoiding extensive signal post-processing but focusing on signal comparison after ensuring three main aspects: I) comparing resting state EEG; II) excluding bad channels; III) selecting artifact-free data for the analysis. The selected analysis approach ensures generalizability and ease of interpretation of the results in the light of future mission designs.

Our analysis results provide evidence for the applicability of EEG in neurocognitive assessment during spaceflight. EEG recorded with unshielded cabling aboard ISS provides sufficient signal quality, with comparable or lower noise levels compared to unshielded ground-level recordings. Signal quality may be further improved by implementation of techniques for environmental noise level reduction, e.g., active shielding. Moreover, multichannel EEG during long-duration space missions may be fostered by the application of dry electrodes providing equivalent signal characteristics when compared to gel-based systems.

The findings of previous neurocognitive EEG studies along with the results of our analyses, and the advantageous characteristics of state-of-the-art mobile dry EEG systems, therefore, underline both the importance and the applicability of in-flight multichannel EEG monitoring in spacecrafts. Consequently, EEG monitoring has the potential to become an important tool in periodical neurocognitive assessment for investigation, diagnostic, and therapeutic support during future long-term spaceflight missions.

## Author Contributions

**Conceptualization:** Patrique Fiedler, Jens Haueisen, Fernando Maestú, Michael Funke.

**Data curation:** Ana M. Cebolla Alvarez, Pablo Cuesta.

**Formal analysis:** Patrique Fiedler.

**Investigation:** Patrique Fiedler.

**Methodology:** Patrique Fiedler, Jens Haueisen.

**Resources:** Patrique Fiedler, Jens Haueisen, Ana M. Cebolla Alvarez, Guy Cheron, Fernando Maestú, Michael Funke.

**Visualization:** Patrique Fiedler.

**Writing – original draft:** Patrique Fiedler, Jens Haueisen.

**Writing – review & editing:** Patrique Fiedler, Jens Haueisen, Ana M. Cebolla Alvarez, Guy Cheron, Pablo Cuesta, Fernando Maestú, Michael Funke.

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
