## [Decision Letter · Decision Letter 0]

10 Jan 2023

Noise characteristics in spaceflight multichannel EEG

PONE-D-22-29775

Dear Dr. Fielder,

We’re pleased to inform you that your manuscript has been judged scientifically suitable for publication and will be formally accepted for publication once it meets all outstanding technical requirements.

Kind regards,

Mukesh Dhamala, Ph. D.

Academic Editor

PLOS ONE

   "This work was supported in part by the German Federal Ministry of Education and Research (BMBF) grant TeleBrain (01DS19009A, Jens Haueisen), the Free State of Thuringia within the ThiMEDOP project (2018 IZN 0004, Jens Haueisen) with funds of the European Union (EFRE), the Belgian Federal Science Policy Office and the European Space Agency (ESA) (AO2004, 118, Guy Cheron + Ana M. Cebolla), and the European Union’s Horizon 2020 research and innovation program under a Marie Skłodowska-Curie grant (101007521, Jens Haueisen and Fernando Maestú)."

Please respond by return e-mail so that we can amend your financial disclosure and competing interests on your behalf.

Additional Editor Comments (optional):

This study of spaceflight noise characteristics of scalp EEG is well-written, well-designed. The comparative analyses of EEG signals inflight to other scenarios confirm that reliable EEG signals can be recorded in spaceflights. It can be published as it is.

Reviewers' comments:

Reviewer's Responses to Questions

**Comments to the Author**

1. Is the manuscript technically sound, and do the data support the conclusions?

Reviewer #1: Yes

2. Has the statistical analysis been performed appropriately and rigorously? 

Reviewer #1: Yes

3. Have the authors made all data underlying the findings in their manuscript fully available?

Reviewer #1: Yes

4. Is the manuscript presented in an intelligible fashion and written in standard English?

Reviewer #1: Yes

5. Review Comments to the Author

Reviewer #1: The authors of this work investigated the noise levels in EEG recordings in space flights for different recording conditions. This manuscript is clearly written and all the results and analysis are well presented. This manuscript can be published in its current form.

6. PLOS authors have the option to publish the peer review history of their article (what does this mean?). If published, this will include your full peer review and any attached files.

Reviewer #1: No

---

## [Editor Report · Acceptance letter]

9 Feb 2023

PONE-D-22-29775 

Noise characteristics in spaceflight multichannel EEG 

Dear Dr. Fiedler:

I'm pleased to inform you that your manuscript has been deemed suitable for publication in PLOS ONE. Congratulations! Your manuscript is now with our production department. 

Kind regards, 

on behalf of

Dr. Mukesh Dhamala 

Academic Editor

PLOS ONE